# Multi-Resolution Weak Supervision
# for Sequential Data

**Frederic Sala**\*    **Paroma Varma**\*    **Jason Fries**    **Daniel Y. Fu**
**Shiori Sagawa**    **Saelig Khattar**    **Ashwini Ramamoorthy**
**Ke Xiao**    **Kayvon Fatahalian**    **James Priest**    **Christopher Ré**
{fredsala, paroma, jfries, danfu, sagawas, saelig, ashwinir, ˙
k̇ayvonf, jpriest, chrismre}@stanford.edu, kexiao@cs.umass.edu

## Abstract

Since manually labeling training data is slow and expensive, recent industrial and scientific research efforts have turned to *weaker* or noisier forms of supervision sources. However, existing weak supervision approaches fail to model *multi-resolution* sources for sequential data, like video, that can assign labels to individual elements or collections of elements in a sequence. A key challenge in weak supervision is estimating the unknown accuracies and correlations of these sources without using labeled data. Multi-resolution sources exacerbate this challenge due to complex correlations and sample complexity that scales in the length of the sequence. We propose Dugong, the first framework to model multi-resolution weak supervision sources with complex correlations to assign probabilistic labels to training data. Theoretically, we prove that Dugong, under mild conditions, can uniquely recover the unobserved accuracy and correlation parameters and use parameter sharing to improve sample complexity. Our method assigns *clinician-validated* labels to population-scale biomedical video repositories, helping outperform traditional supervision by 36.8 F1 points and addressing a key use case where machine learning has been severely limited by the lack of expert labeled data. On average, Dugong improves over traditional supervision by 16.0 F1 points and existing weak supervision approaches by 24.2 F1 points across several video and sensor classification tasks.

## 1   Introduction

Modern machine learning models rely on a large amount of labeled data for their success. However, since hand-labeling training sets is slow and expensive, domain experts are turning to *weaker*, or noisier forms of supervision sources like heuristic patterns [10], distant supervision [18], and user-defined programmatic functions [22] to generate training labels. The goal of *weak supervision* frameworks is to automatically generate training labels to supervise arbitrary machine learning models by estimating unknown source accuracies [8, 13, 21, 24, 28, 29].

Using these frameworks, practitioners can leverage the power of complex, discriminative models without hand-labeling large training sets by encoding domain knowledge in supervision sources. This approach has achieved state-of-the-art performance in many applications [19, 28] and has been deployed by several large companies [2, 4, 5, 11, 15, 16]. However, current techniques do not account for sources that assign labels at *multiple resolutions* (e.g. labeling individual elements and collections of elements), which is common in sequential modalities like sensor data and video.

Consider training a deep learning model to detect interviews in TV news videos [7]. As shown in Figure 1, supervision sources used to generate *training labels* can draw on indirect signals from closed caption transcripts (per-scene), bounding box movement between frames (per-window), and pixels in the background of each frame (per-frame). However, existing weak supervision frameworks cannot

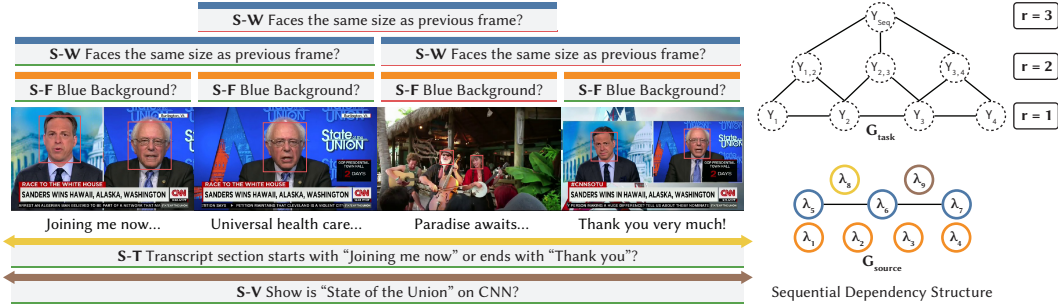

Figure 1: Multi-resolution weak supervision sources to label video analytics training data. S-X outputs noisy label vectors $\lambda_j$ and represents various supervision sources at different resolutions: Video (S-V), Transcript (S-T), Window (S-W), and Frame (S-F) (brown, yellow, blue, orange). We show a graphical model structure for modeling these sources at different resolutions ($r = 1,2,3$): dotted nodes represent latent true labels, solid nodes represent the noisy supervision sources, and edges represent sequential relations.

model two key aspects of this style of sequential supervision. First, sources are *multi-resolution* and can assign labels on a per-frame to per-window to per-scene basis, implicitly creating *sequential correlations* among the noisy supervision sources that can lead to conflicts within and across resolutions. Second, we have no principled way to incorporate *distribution prior*, like how frames with interviews are distributed within a scene—and this is critical for temporal applications.

The core technical challenge in this setting is integrating diverse sources with unknown correlations and accuracies at scale *without observing any ground truth labels*. Traditionally, such issues have been tackled via probabilistic graphical models, which are expressive enough to capture sequential correlations in data. Unfortunately, learning such models via classical approaches such as variational inference [27] or Gibbs sampling [14] presents both practical and theoretical challenges: these techniques often fail to scale, in particular in the case of long sequences. Moreover, algorithms for latent-variable models may not always converge to a unique solution, especially in cases with complex correlations.

We propose Dugong— the first weak supervision framework to integrate multi-resolution supervision sources of varying quality and incorporate distribution prior to generate high-quality training labels. Our model uses the agreements and disagreements among diverse supervision sources, instead of traditional hand-labeled data, at different resolutions (e.g., frame, window, and scene-level) to output probabilistic *training labels* at the required resolution for a downstream end model. We develop a simple and scalable approach that estimates parameters associated with source accuracy and correlation by solving a pair of linear systems.

We develop conditions under which the underlying statistical model is identifiable. With mild conditions on the correlation structure of sources, we prove that the model parameters are recoverable directly from the systems. We show that we can reduce the dependence of sample complexity on the length of the sequence from exponential to linear to independent, using various degrees of parameter sharing, which we analyze theoretically. Applying recent results in weak supervision literature, we then show that the generalization error of the end model scales as $O(1/\sqrt{n})$ in the number of unlabeled data points—the same asymptotic rate as supervised approaches.

We experimentally validate our framework on five real-world sequential classification tasks over modalities like medical video, gait sensor data, and industry-scale video data. For these tasks, we collaborate with domain experts like cardiologists to create multi-resolution weak supervision sources. Our approach outperforms traditional supervision by $16.0$ F1 points and existing state-of-the-art weak supervision approaches by $24.2$ F1 points on average.

We also create an SGD variant of our method that enables implementation in modern frameworks like PyTorch and achieves $90\times$ faster runtimes compared to prior Gibbs-sampling based approaches [1, 22]. This scalability enables using clinician-generated supervision sources to automatically label population-scale biomedical repositories such as the UK Biobank [23] on the order of days, addressing a key use case where machine learning has been severely limited by the lack of expert labeled data and improving over state-of-the-art traditional supervision by $31.7$ F1 points.

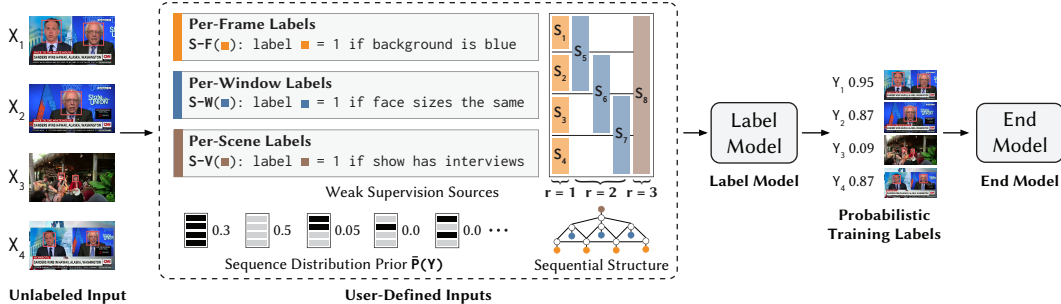

Figure 2: A schematic of the `Dugong` pipeline. Users provide a set of unlabeled sequences where each sequence $X = [X_1,...,X_T]$, a set of weak supervision sources $S_1,...,S_m$, each of which assigns labels at multiple resolutions (frame, window, scene), a sequential structure (i.e., $G_{\text{source}}$ and $G_{\text{task}}$), and a distribution prior $\bar{P}_Y$. The label model estimates the unknown accuracies and correlation strengths of the supervision sources and assigns probabilistic training labels to each element, which can be used to train a downstream end model.

## 2 Training Machine Learning Models with Weak Supervision

Practitioners often weakly supervise machine learning models by programmatically generating training labels through the process shown in Figure 2. First, users provide multiple *weak supervision sources*, which assign noisy labels to unlabeled data. These labels overlap and conflict, and a *label model* is used to integrate them into probabilistic labels. These probabilistic labels are then used to train a discriminative model, which we refer to as the end model.

While generating training labels across various sequential applications, we found that supervision sources often assign labels at *different resolutions*: given a sequence with $T$ elements, sources can assign a single label per element, per collection of elements, or for the entire sequence. We describe a set of such supervision sources as *multi-resolution*. For example in Figure 1, to train an end model that detects interviews in TV shows, noisy labels can be assigned to each frame, each window, or each scene. Sources `S-F`, `S-W`, and `S-V` each assign labels to a frame at resolution level $r = 1$, a window at $r = 2$, and scene at $r = 3$, respectively. While each source operates at a specific resolution, the sources together are multi-resolution. The main challenge is combining source labels into probabilistic training labels by estimating source accuracies and correlations without ground-truth labels.

### 2.1 Problem Setup

We set up our classification problem as follows:

- Let $X = [X_1, X_2,...,X_T] \in \mathcal{X}$ be an unlabeled sequence with $T$ elements (video frames in Figure 1).

- For each sequence $X$, we assign labels to tasks at multiple resolutions ($Y_1$, $Y_{1,2}$, $Y_{seq}$ etc. in Figure 1). We formally refer to the tasks using indices $\mathcal{T} = \{1,...,|\mathcal{T}|\}$ ($|\mathcal{T}| = 4+3+1 = 8$ for the resolutions $r = 1,2,3$ shown in Figure 1).

- These tasks are at multiple resolutions (3 resolutions in Figure 1) with the set of tasks at resolution $r$ denoted $R_r \subseteq \mathcal{T}$.

- $Y \in \mathcal{Y}$ is a vector $[y_1,...,y_{|\mathcal{T}|}]$ of unobserved true labels for each task, and $(X,Y)$ are drawn i.i.d. from some distribution $\mathcal{D}$.

Users provide $m$ multi-resolution sources $S_1,...,S_m$. Each source $S_j$ assign labels $\lambda_j$ to a set of tasks $\tau_j \subseteq \mathcal{T}$, (henceforth *coverage set*), with size $s_j = |\tau_j|$. Each source only assigns labels at a specific resolution $r$, enforcing $\tau_j \subseteq R_r$ for fixed $r$. Users also provide a *task dependency graph* $G_{\text{task}}$ specifying relations among tasks, a *source dependency graph* $G_{\text{source}}$ specifying relations among supervision sources that arise due to shared inputs (Figure 1), and a *distribution prior* $\bar{P}(Y)$ describing likelihood of labels in a sequence (Figure 2). While $G_{\text{source}}$ is user-defined, it can also be learned directly from the source outputs [1, 26].

We want to apply weak supervision sources $S$ to an unlabeled dataset $X$ consisting of $n$ sequences, combine them into probabilistic labels, and use those to supervise an end model $f_w : \mathcal{X} \rightarrow \mathcal{Y}$ (Figure 2). Since the labels from the supervision sources overlap and conflict, we learn a label model $P(Y|\lambda)$ that takes as input the noisy labels and outputs probabilistic labels *at the resolutions required* by the end model.

## 2.2   Label Model

Given inputs $X, S, G_{\text{task}}, G_{\text{source}}, \bar{P}(Y)$, we estimate the sources' unknown accuracies and correlation strengths. Accuracy parameters $\mu$ and correlation parameters $\phi$ define a *label model* $P_{\mu,\phi}(Y|\lambda)$, which can generate probabilistic training labels. To recover parameters without ground-truth labels $Y$, we observe the agreements and disagreements of these noisy sources across different resolutions.

To recover these parameters, we form a graph $G$ describing all relations among sources and task labels, combining $G_{\text{source}}$ with $G_{\text{task}}$. The resulting graphical model encodes conditional independence structures. Specifically, if $(\lambda_j, \lambda_k)$ is not an edge in $G$, then $\lambda_j$ and $\lambda_k$ are independent conditioned on all of the other variables.

For ease of exposition, we assume the binary classification setting where $y_i \in \{-1, 1\}$, $\lambda_i \in \{-1, 1\}$ for $T$ per-element tasks and 1 per-sequence task. The accuracy parameter for source $j$ for some $Z, W \in \{-1, 1\}^{s_j + 1}$ is

$$\mu_j(Z, W) = P\left(\lambda_j = Z \mid Y_{\tau_j} = W\right). \tag{1}$$

Intuitively, this parameter captures the *accuracy* of each supervision source with respect to the ground truth labels in coverage set $\tau_j$. Next, for each correlation pair of sources $(\lambda_j, \lambda_k)$ and for some $Z_1 \in \{-1, 1\}^{s_j}, Z_2 \in \{-1, 1\}^{s_k}, W \in \{-1, 1\}^{|\tau_j \cup \tau_k|}$, we wish to learn

$$\phi_{j,k}(Z_1, Z_2, W) = P(\lambda_j = Z_1, \lambda_k = Z_2 \mid Y_\tau = W), \tag{2}$$

where $\tau = \tau_j \cup \tau_k$.

## 2.3   Parameter Reduction

Our assumption above of conditioning only on ground-truth labels for tasks in the source's coverage set instead of the full $\mathcal{T}$ greatly reduces the number of parameters. While we have at least $2^T$ parameters without the assumption, we now only need to learn $2^{2s_j}$ parameters per source, where $s_j$ tends to be much smaller than $T$.

In addition, we can model each source accuracy conditioned on each task, rather than over its full coverage set, reducing from $2^{2s_j}$ to $4s_j$ parameters and going from exponential to linear dependence on coverage set size, which is at most $T$. Lastly, we can also use parameter sharing: we share across sources that apply the same logic to label different, same-resolution tasks ($\mu_1 = \mu_2 = \mu_3 = \mu_4$ in Figure 1).

# 3   Modeling Sequential Weak Supervision

The key challenge in sequential weak supervision settings is recovering the unknown accuracies and correlation strengths in our graphical model of multi-resolution sources, given the noisy labels, the dependency structures $G_{\text{source}}$ and $G_{\text{task}}$, coverage sets $\tau$, and distribution prior $\bar{P}_Y$. We propose a provable algorithm that recovers the unique parameters with convergence guarantees by reducing parameter recovery into systems of linear equations. These systems recover probability terms that involve the unobserved true label $Y$ by exploiting the pattern of agreement and disagreement among the noisy supervision sources at different levels of resolution (Section 3.1). We theoretically analyze this algorithm, showing how the estimation error scales with the number of samples $n$, the number of sources $m$, and the length of the sequence $T$. Our approach additionally leverages repeated structures in sequential data by sharing appropriate parameters, significantly reducing sample complexity to no more than linear in the sequence length (Section 3.2). Finally, we consider the impact of our estimation error on the end model trained with labels produced from our label model, showing that end model generalization scales with unlabeled data points as $O(1/\sqrt{n})$, the same asymptotic rate as if we had access to labeled data (Section 3.2).

## 3.1 Source Accuracy Estimation Algorithm

Our approach is shown in Algorithm 1: it takes as input samples of sources $\lambda_1,...,\lambda_m$, independencies resulting from the graph $G$, and the prior $\bar{P}_Y$ and outputs the estimated accuracy and correlation parameters, $\hat{\mu}$ and $\hat{\phi}$ (for simplicity, we only show the steps for $\mu$ in Algorithm 1.)

While we have access to the noisy labels assigned by the supervision sources, we do not observe the true labels $Y$ and therefore cannot calculate $\mu$ directly. However, given access to the user-defined distribution prior and the *joint probabilities*, such as $P(\lambda_j(\{1\}),y_2)$, we can apply Bayes' law to estimate $\mu$ (Section 3.1.4). Since the joint probabilities also include the unobservable $Y$ term, we break it into the sum of *product variables*, such as $P(\lambda_j(\{1\})y_2=1)$ (Section 3.1.3). Note that we still have a dependence on the true label $Y$: to address this issue, we take advantage of (1) the conditional independence of some sources (Section 3.1.2), (2) the fact that we can observe the agreement and disagreements among the sources (Section 3.1.1), and (3) in the binary setting, $y^2=1$.

We describe the steps of our algorithm and explain the assumptions we require, which involve the number of conditionally independent pairs of sources we have access to and how accurately they vote on their tasks.

---

**Algorithm 1:** Accuracy Parameter Estimation

**Input:** Samples of sources $\lambda_1,...,\lambda_n$, Dependency structure $G$, Dist. prior $\bar{P}(Y)$

1   **for** *source* $j \in \{1,...,m\}$ **do**
2     **for** *coverage subsets* $U,V \subseteq \tau_j$ **do**
3       Using $G$, get source set $S_j$ where $\forall k,\ell \in S_j, \exists U_k,U_\ell$
        s.t. $a_j(U,V) \perp a_k(U_k,V)$, $a_j(U,V) \perp a_\ell(U_\ell,V)$, $a_k(U_k,V) \perp a_\ell(U_\ell,V)$. Set $U_j = U$
4       **for** $k,\ell \in S_j \cup \{j\}$ **do**
5         Calculate **gen. agreement measure**: $a_k(U_k,V)a_\ell(U_\ell,V) = \prod_{U_k,U_\ell} \lambda_k(U_k)\lambda_\ell(U_\ell)$
6         Form $q = \log \mathbb{E}[a_k(U_k,V)a_\ell(U_\ell,V)]^2$ over coverage subsets $U_k,U_\ell,V$
7       Solve **agreement-to-products** system: find $\ell_{U,V}$ s.t. $M\ell_{U,V} = q$
8     Form product probability vector $r(\ell_{U,V})$
9     Solve **products-to-joints** system: find $e$ s.t. $B_{2s_j}e = r$
10    $\mu_j \leftarrow e/\bar{P}(Y)$

**Output :** Parameter $\hat{\mu}$

---

### 3.1.1 Generalized Agreement Measure

Given the noisy labels assigned by the supervision sources, $\lambda_1,...,\lambda_m$, we want some measure of agreement between these sources and the true label $Y$. For sources $j$ and $k$, let $U,U',V$ be subvectors of the coverage sets $\tau_j,\tau_k,\tau_j \cup \tau_k$, respectively. We use the notation $\prod_X A(X)$ to represent the product of all components of $A$ indexed by $X$. We then define a *generalized agreement measure* as $a_j(U,V) = \prod \lambda_j(U)\prod Y(V)$, which represents the agreement between the supervision source and the unknown true label when $U=V$ and $|U|=1$. Note that this term is *not directly observable as it is a function of $Y$*.

Instead, we look at the product of two such terms:

$$a_j(U,V)a_k(U',V) = \prod_{U,U'} \lambda_j(U)\lambda_k(U') \prod_V (Y(V))^2 = \prod_{U,U'} \lambda_j(U)\lambda_k(U').$$

Since the $(Y(V))^2$ components multiply to 1 in the binary setting, we are able to represent the product of two generalized agreement measures in terms of the *observable agreement and disagreement between supervision sources*. Therefore, we are able to calculate $a_j(U,V)a_k(U',V)$ across values of $U,V$ directly from the observed variables.

### 3.1.2 Agreements-to-Products System

Given the product of generalized agreement measures, we solve for terms that involve the true label $Y$, such as $a_j(U,V)$. Since we cannot observe these terms directly, we instead solve a system of equations that involve $\log \mathbb{E}[a_j(U,V)]$, the *log of the expectation of these values* when we have certain *assumptions about independence* of different sources, conditioned on variables from $Y$. We give more

details in the Appendix. As an example, note that if $\lambda_j(U)$ is independent of $\lambda_k(U')$ given $\prod Y(V)$ for $|V| = 1$, which is information that can be read off of the graphical model $G$, then

$$\mathbb{E}[a_j(U,V)]\mathbb{E}[a_k(U',V)] = \mathbb{E}[a_j(U,V)a_k(U',V)] = \mathbb{E}\Big[\prod_{U,U'} \lambda_j(U)\lambda_k(U')\Big]. \qquad (3)$$

In other words, the conditional independencies of the sources translate to independencies of the accuracy-like terms $a$.

Note that the middle term in (3) can be calculated directly using observed $\lambda$'s. Now we wish to form a system of equations to solve for the terms on the left-most side. We can take the log of the left-most term and the right-most term to form a system of linear equations, $M\ell = q$. $M$ contains a row for each pair of sources, $\ell$ is the vector we want to solve for and contains the terms with $a_j(U,V)$, and $q$ is the vector we observe and contains the terms with $\lambda_j(U)\lambda_k(U')$. We can solve this system up to sign if $M$ is full rank, which is true if $M$ has at least three rows. This is true if we have a group of at least three conditionally independent sources.

**Assumptions** We now have the notation to formally state our assumptions. We assume that each $a_j(U,V)$ has at least two other independent accuracies (equivalently, sources independent conditioned on $Y_V$) and $|\mathbb{E}[a_j(U,V)]| > 0$, i.e., our accuracies are correlated with our labels, positively or negatively), and that we have a list of such independencies (to see how to obtain such a list from the user-provided graphs, more information is in the Appendix). We also assume that on average, a group of connected sources have a better than random chance of agreeing with the labels, which enables us to recover the signs of the accuracies. These are standard weak supervision assumptions [20].

Once we solve for $\mathbb{E}[a_j(U,V)]$, we can calculate the *product variable* probabilities $\rho_j(U, V) = P(a_j(U, V) = 1) = 1/2(1 + \mathbb{E}[a_j(U,V)])$. Note that product variable probabilities $\rho$ relies on the the true label $Y$, since $a_j(U,V)$ represents the generalized agreement between the source label and true label. However, we have now solved for this term *despite not observing $Y$ directly*.

### 3.1.3 Products-to-Joints System

Given the product variable probabilities, we now want to solve for the *joint probabilities* $p$, such as $P(\lambda_{j,1},Y_2)$. Fortunately, linear combinations of the appropriate $p_j(Z,W) = P(\lambda_j = Z, Y_{\tau_j} = W)$ result in $\rho_j(U,V)$ terms. Our goal is to solve for the unknown joint probabilities given the estimated $\rho_j$ product variables, user-defined distribution prior $\bar{P}_Y$, and observed labels from the sources $\lambda$.

Say that $\lambda_1$ has coverage $\tau_1 = [1]$, so that it only votes on the value of $y_1$. Then, for $U = \{1\}, V = \{1\}$, we know $\rho_1(U,V) = P(\lambda_{1,1}y_1 = 1)$. But we have that $P(\lambda_{1,1}y_1 = 1) = p_1(+1,+1) + p_1(-1,-1)$, which is the agreement probability. Using similar logic, we can set up a series of linear equations:

$$\begin{bmatrix} 1 & 1 & 1 & 1 \\ 1 & 0 & 1 & 0 \\ 1 & 1 & 0 & 0 \\ 1 & 0 & 0 & 1 \end{bmatrix} \begin{bmatrix} p_1(+1,+1) \\ p_1(-1,+1) \\ p_1(+1,-1) \\ p_1(-1,-1) \end{bmatrix} = \begin{bmatrix} 1 \\ P(\lambda_{1,1}=1) \\ P(Y_1=1) \\ \rho_1(U,V) \end{bmatrix}.$$

Note that because of how we set up this system, the vector on the left-hand side contains the probabilities we need to estimate the joint probabilities. The right hand side vector contains either observable ($P(\lambda_{1,1}=1)$), estimated ($\rho_1(U,V)$), or user-defined ($P(Y_1=1)$, from $\bar{P}_Y$) terms. In this example, the matrix is full-rank and we can therefore solve for the $p_1$ terms.

To extend this system to the general case, we form a system of linear equations, $B_{2s_j}e = r$. $B_{2s_j}$ is the *products-to-joints* matrix (we discuss its form below), $e$ is the vector we want to solve for and contains the $p_j(Z,W)$ terms, and $r$ is the vector we have access to and contains observable, user-defined, and estimated $\rho_j(U,V)$ terms. $B_{2s_j}$ is $2^{2s_j} \times 2^{2s_j}$-dimensional 0/1 matrix. Let $\otimes$ be the Kronecker product; then, we can represent $B_{2s_j}$ as a Hadamard-like matrix (we show it is full rank in the Appendix):

$$B_{2s_j} = \frac{1}{2}\begin{bmatrix} 1 & 1 \\ 1 & -1 \end{bmatrix}^{\otimes 2s_j} + \frac{1}{2}11^T.$$

We can now solve for terms required to calculate the joint probabilities and use them to obtain the $\mu$ parameters by using Bayes' law and the user-defined distribution prior $\mu_j(z,w) = p_j(Z,W)/P(Y_{\tau_j} = W)$. We can calculate the $\phi$ parameters in a similar fashion as $\mu$, except now we operate over *pairs* of supervision sources, always working with products of correlated sources $\lambda_i\lambda_j$ (details in Appendix).

### 3.1.4 SGD-Based Variant

Note that Algorithm 1 explicitly builds and solves the linear systems that are set up via the agreement measure constraints. This involves a small amount of bookkeeping. However, there is a simple variant that relies on SGD for optimization and simply uses the constraints between the accuracies and correlations. That is, we use $\ell_2$ losses on the constraints (and additional ones required to make the probabilities consistent) and directly optimize over the accuracy and correlation variables $\mu, \phi$. Under the assumptions we have set up in this section, these algorithms are effectively equivalent; in the experiments, we use the SGD-based variant due to its ease of implementation in PyTorch.

### 3.2 Theoretical Analysis: Scaling with Sequential Supervision

Our ultimate goal is to train an end model using the labels aggregated from the supervision sources using the estimated $\mu$ and $\phi$ for the label model. We first analyze Algorithm 1 with parameter sharing as described in Section 2.3 and discuss the general case in the Appendix. We bound our estimation error and observe the scaling in terms of the number of unlabeled samples $n$, the number of sources $m$, and the length of our sequence $T$. We then connect the generalization error to the end model to the estimation error of Algorithm 1, showing that generalization error scales asymptotically in $O(\sqrt{1/n})$, the same rate as supervised methods but in terms of number of *unlabeled* sequences.

We have $n$ samples of each of the $m$ sources for sequences of length $T$, and the graph structure $G = (V, E)$. We allow for coverage sets of size up to $T$. We assume the previously-stated conditions on the availability of conditionally independent sources are met, that $\forall j, |\mathbb{E}[a_j(U, V)]| \geq b_{\min}^* > 0$, and that sign recovery is possible (for example, it is sufficient to have $\forall j, U, V, \sum_{\lambda_k \in S_j} \mathbb{E}[a_k(U, V)] > 0$ where $S_j$ is defined as in Algorithm 1). We also take $p_{\min}$ to be the smallest of the entries in $\bar{P}(Y)$. Let $\|\cdot\|$ be the spectral norm.

**Theorem 1** *Under the assumptions above, let $\hat{\mu}$ and $\hat{\phi}$ be estimates of the true $\mu^*$ and $\phi^*$ produced with Algorithm 1 with parameter reduction. Then,*

$$\mathbb{E}[\|\hat{\mu} - \mu^*\|] \leq \sqrt{mT} \frac{24}{p_{\min} b_{\min}^*} \|B_{2T}^{-1}\| \|M^\dagger\| \left( \sqrt{\frac{18 \log(12)}{n}} + \frac{2 \log(12)}{n} \right). \tag{4}$$

*The expectation $\mathbb{E}[\|\hat{\phi} - \phi^*\|]$ satisfies the bound* (4)*, replacing $\sqrt{mT}$ with $mT$ and $B_2$ with $B_4$.*

**Interpreting the Theorem** The above formula scales with $n$ as $O(\sqrt{1/n})$, and critically, *no more than linear in $T$*. We prove a more general bound without parameter reduction, which scales exponentially in $T$ in the Appendix. The expression scales with $m$ as $O(\sqrt{m})$ and $O(m)$ for estimating $\mu$ and $\phi$, respectively. The standard scaling factors for the random vectors produced by the sources are $m$ and $m^2$; however, we need *only two additional sources for each source*, leading to the $\sqrt{m}$ and $m$ rates. The linear systems enter the expression only via $\|B^\dagger\|$. These are fixed; in particular, $\|B_2^\dagger\| = 1.366$ and $\|B_4^\dagger\| = 1.112$.

**End Model Generalization** After obtaining the label model parameters, we use them to generate probabilistic training labels for the resolution required by the end model. The parameter error bounds from Theorem 1 allow us to apply a result from [20], which states that under the common weak supervision assumptions (e.g., the parameters of the distribution we seek to learn are in the space of the true distribution), the generalization error for $Y$ satisfies $\mathbb{E}[l(\hat{w}, X, Y) - l(w^*, X, Y)] \leq \gamma + 8(\|\hat{\mu} - \mu^*\| + \|\hat{\phi} - \phi^*\|)$. Here, $l$ is a bounded loss function and $w$ are the parameters of an end model $f_w : \mathcal{X} \to \mathcal{Y}$. We also have $\hat{w}$ as the parameters learned with the estimated label model using $\mu$ and $\phi$, and $w^* = \text{argmin}_w l(w, X, Y)$, the minimum in the supervised case. This result states that the generalization error for our end models *scales with the amount of unlabeled data as $O(1/\sqrt{n})$*, the same asymptotic rate as if we had access to the true labels.

## 4 Experimental Results

We validate `Dugong` on real-world sequential classification problems, comparing end model performance trained on labels from `Dugong` and other baselines. `Dugong` improves over traditional

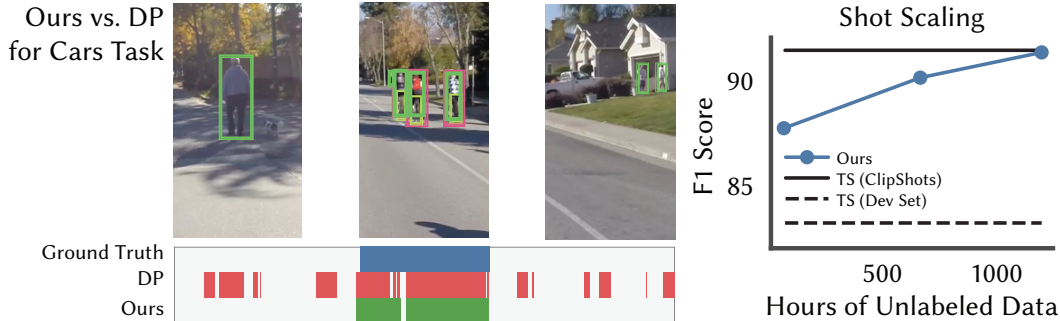

Figure 3: (Left) Dugong has fewer false positives than data programming on a cyclist detection task since it uses sequential correlations and distributional knowledge to assign better training labels. (Right) Increasing unlabeled data can help match a benchmark model trained with $686\times$ more ground truth data, i.e., using traditional supervision (TS).

supervision and other state-of-the-art weak supervision methods by $16.0$ and $24.2$ F1 points on average in terms of end model performance, respectively. We also conduct ablations to compare parameter reduction techniques, the effect of modeling dependencies, and the advantage of using a user-defined prior, with average improvements of $3.7$, $10.4$, and $13.7$ F1 points, respectively. Finally, we show how our model scales with the amount of unlabeled data, coming within $0.1$ F1 points of a model trained on $686\times$ more ground-truth labels.

## 4.1 Datasets

We consider two types of tasks, spanning various modalities: (a) tasks that are expensive and slow to label due to the domain expertise required, and (b) previously studied, large-scale tasks with strong baselines often based on hand-labeled data developed over months. All datasets include a small hand-labeled *development set* ($< 10\%$ of the unlabeled data) used to tune supervision sources and end model hyperparameters. Results are reported on test set as the mean $\pm$ S.D. of F1 scores across 5 random weight initializations. See Appendix for additional task and dataset details, precision and recall scores, and end model architectures.

**Domain Expertise** These tasks can require hours of expensive expert annotations to build large-scale training sets. The Bicuspid Aortic Valve **(BAV)** [6] task is to classify a congenital heart defect over MRI videos from a population-scale dataset [23]. Labels generated from Dugong and sources based on characteristics like heart area and perimeter are *validated by cardiologists*. Interview Detection **(Interview)** identifies interviews of Bernie Sanders from TV news broadcasts; across a large corpus of TV news, interviews with Sanders are rare, so it requires significant labeling effort to curate a training set. Freezing Gait **(Gait)** is ankle sensor data from Parkinson's patients and the task is to classify abnormal gait [12], using supervision sources over characteristics like peak-to-peak distance. Finally, **EHR** consists of tagging mentions of disorders in patient notes from electronic health records. We only report label model results for **EHR**, but Dugong improves over a majority vote baseline by $3.7$ F1 points (see Appendix).

**Large-Scale** Movie Shot Detection **(Movie)** classifies frames that contain a change in scene using sources that use information about pixel values, frame-level metadata, and sequence-level changes. This task is well-studied in literature [9, 25] but adapting the method to specialized videos requires manually labeling thousands of minutes of video. Instead, we use $686\times$ fewer ground truth labels and various supervision sources to *match the performance of a model pre-trained on a benchmark dataset with ground truth labels* (Figure 3). **Basketball** operates over a subset of ActivityNet [3] and uses supervision sources over frames and sequences. Finally, we use a representative dataset for cyclist detection from a large automated driving company **(Cars)** [17] and show that *we outperform their best baseline by 9.9 F1 points*. The **Cars** end model is proprietary, so we only report label model results (Appendix).

| Task | Prop | T | End Model Performance | | | | Improvement | | |
|---|---|---|---|---|---|---|---|---|---|
| | | | TS | MV | DP | Dugong | TS | MV | DP |
| BAV | 0.07 | 5 | $22.1 \pm 5.1$ | $6.2 \pm 7.6$ | $53.2 \pm 4.4$ | $\mathbf{53.8 \pm 7.6}$ | +31.7 | +47.6 | +0.6 |
| Interview | 0.03 | 5 | $80.0 \pm 3.4$ | $58.0 \pm 5.3$ | $8.7 \pm 0.2$ | $\mathbf{92.0 \pm 2.2}$ | +12.0 | +34.0 | +83.3 |
| Gait | 0.33 | 5 | $47.5 \pm 14.9$ | $61.6 \pm 0.4$ | $62.9 \pm 0.7$ | $\mathbf{68.0 \pm 0.7}$ | +20.5 | +6.4 | +5.1 |
| Shot | 0.10 | 5 | $83.2 \pm 1.0$ | $86.0 \pm 0.9$ | $86.2 \pm 1.1$ | $\mathbf{87.7 \pm 1.0}$ | +4.5 | +1.7 | +1.5 |
| Basketball | 0.12 | 5 | $26.8 \pm 1.3$ | $8.1 \pm 5.4$ | $7.7 \pm 3.3$ | $\mathbf{38.2 \pm 4.1}$ | +11.4 | +30.1 | +30.5 |

Table 1: End model performance in terms of F1 score (mean $\pm$ std.dev). Improvement in terms of mean F1 score. Prop: proportion of positive examples in the dev set, $T$: number of elements in a sequence. We compare end model performance on labels from labeled dev set (TS), majority vote across sources (MV), and data programming (DP) and outperform each across all tasks.

## 4.2 Baselines

For the tasks described above, we compare to the following baselines (Table 1): *Traditional Supervision (TS)* in which end models are trained using the hand-labeled development set; *Non-sequential Majority Vote (MV):* in which we force all supervision sources assign labels per-element, and calculate training labels by taking majority vote across sources; and *Data Programming (DP) [21]:* a state-of-the-art weak supervision technique that learns the accuracies of the sources but does not model sequential correlations.

In tasks with domain expertise required, our approach improves over traditional supervision by up to 36.8 F1 points and continually improves precision as we add unlabeled data, as shown in Figure 3. Large-scale datasets have manually curated baselines developed over *months*; Dugong is still able to improve over baselines by up to 30.5 F1 points by capturing sequential relations properly — as shown in Figure 3, only modeling source accuracies (DP) can fail to take into account the distribution prior and sequential correlations among sources that can help filter false positives, which Dugong does successfully.

## 4.3 Ablations

We demonstrate how each component of our model is critical by comparing end model performance trained on labels from Dugong without any sequential dependencies, Dugong without parameter sharing for sources with shared logic (Section 2.3), and Dugong with various distribution priors: user-defined, development-set based, and uniform. We report these comparisons in the Appendix and summarize results here.

Without sequential dependencies, end model performance worsens by 10.4 F1 points on average, highlighting the importance of modeling correlations among sources. We see that sharing parameters among sources that use the same logic to assign labels at the same resolution performs 3.7 F1 points better on average. Using a user-defined distribution prior improves over using a uniform distribution prior by 13.7 F1 points and a development-set based distribution prior by 1.7 F1 points on average, highlighting how domain knowledge in forms other than supervision sources is key to generating high quality training labels.

## 5 Conclusion

We propose Dugong, the first weak supervision framework that integrates multi-resolution weak supervision sources including complex dependency structures to assign probabilistic labels to training sets without using any hand-labeled data. We prove that our approach can uniquely recover the parameters associated with supervision sources under mild conditions, and that the sample complexity of an end model trained using noisy sources matches that of supervised approaches. Experimentally, we demonstrate that Dugong improves over traditional supervision by 16.0 F1 points and existing weak supervision approaches by 24.2 F1 points for real-world classification tasks training over large, population-scale biomedical repositories like UKBiobank [23] and industry-scale video datasets for self-driving cars.

## Acknowledgments

We gratefully acknowledge the support of DARPA under Nos. FA87501720095 (D3M), FA86501827865 (SDH), and FA86501827882 (ASED); NIH under No. U54EB020405 (Mobilize), NSF under Nos. CCF1763315 (Beyond Sparsity), CCF1563078 (Volume to Velocity), and 1937301 (RTML); ONR under No. N000141712266 (Unifying Weak Supervision); the Moore Foundation, NXP, Xilinx, LETI-CEA, Intel, IBM, Microsoft, NEC, Toshiba, TSMC, ARM, Hitachi, BASF, Accenture, Ericsson, Qualcomm, Analog Devices, the Okawa Foundation, American Family Insurance, Google Cloud, Swiss Re, Brown Institute for Media Innovation, the National Science Foundation (NSF) Graduate Research Fellowship under No. DGE-114747, Joseph W. and Hon Mai Goodman Stanford Graduate Fellowship, Department of Defense (DoD) through the National Defense Science and Engineering Graduate Fellowship (NDSEG) Program, and members of the Stanford DAWN project: Teradata, Facebook, Google, Ant Financial, NEC, VMWare, and Infosys. The U.S. Government is authorized to reproduce and distribute reprints for Governmental purposes notwithstanding any copyright notation thereon. Any opinions, findings, and conclusions or recommendations expressed in this material are those of the authors and do not necessarily reflect the views, policies, or endorsements, either expressed or implied, of DARPA, NIH, ONR, or the U.S. Government.

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
