[Supplementary Material]

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

First we discuss related work and then provide a glossary of the terminology and notation used throughout this paper for ease of reference. Afterwards, we provide our theoretical analysis, and extended theorem statement, proofs, and more details on model identifiability. Lastly, we include additional details and experiments.

## A   Related Work

Our work is related to several weak supervision techniques such as traditional distant supervision [8, 23, 38, 51], co-training methods [5], pattern-based supervision [18] and feature annotation techniques [33, 35, 58]. Recent works also use generative models [3, 47, 48] and other methods [17, 29] to integrate these noisy sources. However, these approaches do not handle sequential correlations or multi-resolution sources and require expensive sampling-based techniques that can lead to non-identifiability. One recent approach directly models weak supervision sources using deep generative models for trajectory data, but does not use weak supervision sources to label training data for arbitrary end models [59]. Our proposed approach is also related to recent techniques for estimating classifier accuracies without labeled data in the presence of structural constraints [42]. Similarly, our approach is related to crowdsourcing [10, 27], specifically to spectral and method of moments-based approaches [2, 9, 16, 61]. However, we focus on the settings that are not covered by crowdsourcing, such as multi-resolution sources, sequential correlation structures, and regimes in which a small number of labelers, or sources, assign noisy labels to a large set of datapoints. We also theoretically characterize how the end model trained on labels from noisy sources generalizes.

## B   Glossary

The glossary is given in Table 2 below.

## C   Proofs and Extended Theoretical Analysis

We give more details on the theoretical results we provided in the body. We start by providing the proof of Theorem 1. Afterwards, we discuss model identifiability, expressing tradeoffs involving multi-resolution models. Finally, we provide further detail on simulations and how to access conditional independencies from graphs.

First, we begin with a proof of Theorem 1. The following lemma will be useful. We use a little bit of notation. Let $D = (d_1, \ldots, d_t)$ be a random vector in $\{-1, +1\}^t$. For particular vectors $U, V, Z \in \{-1, +1\}^t$, we write $p_D(Z) = P(d_1 = z_1, \ldots, d_t = z_t)$ and $\rho_D(U) = P(\prod_U D_U = 1)$. The $p$ term is a joint probability and the $\rho$ term is a product probability. Let

$$B_t = \frac{1}{2} \begin{bmatrix} 1 & 1 \\ 1 & -1 \end{bmatrix} \otimes^t + \frac{1}{2} 11^T.$$

Here, $1$ is the all 1's vector and $A \otimes^k$ represents taking the Kronecker product $A \otimes A$ a total of $k$ times.

Let the vector $e$ contain the $2^t$ entries $p_D(Z)$, with $Z$ taken in the following order. $z_t = +1$ for the first $2^{t-1}$ entries and $-1$ for the latter $2^{t-1}$ entries, $z_{t-1} = +1$ for the first $2^{t-2}$ entries, and so on, so that $z_1$ alternates between $+1$ and $-1$. Similarly, let the vector $r$ contain the $2^t$ choices of $\rho_D(U)$ running over all $2^t$ subsets of $\{1, \ldots, t\}$. We write $\rho_D(\emptyset) = 1$. Then, the ordering of the entry in $r$ is similar to the ones in $e$: the first half of the $U$ terms in $\rho_D$ do not contain the entry $t$, the latter half do, and so on, so that every alternating entry contains the entry 1. Then,

**Lemma 1** *With the setup above, $B_t e = r$.*

**Proof:**   We prove the result by induction on $t$. For the base case we take $t = 1$. Then, using the above formula for $B_1$, we must have the following, which is clearly true

$$\begin{bmatrix} 1 & 1 \\ 1 & 0 \end{bmatrix} \begin{bmatrix} p_D([1]) \\ p_D([-1]) \end{bmatrix} = \begin{bmatrix} \rho_D(\emptyset) \\ \rho_D(\{1\}) \end{bmatrix}.$$

| Symbol | Used for |
|---|---|
| $X$ | Unlabeled data sequence, $X = [X_1, X_2, ..., X_T] \in \mathcal{X}$ |
| $T$ | Length of the unlabeled data sequence |
| $n$ | Number of data sequences |
| $\mathcal{T}$ | Task indices |
| $Y$ | Latent, ground-truth label vector, $Y = [y_1, y_2, ..., y_T, y_{T+1}, ..., y_{|\mathcal{T}|}] \in \mathcal{Y}$ |
| $y_i$ | Ground-truth label for $i$th task, $y_i \in \{-1, 1\}$ |
| $\mathcal{D}$ | Distribution from which we assume $(X, Y)$ data points are sampled i.i.d. |
| $r$ | Resolution level. $r = 1$ refers to resolution level in which each of the $T$ elements is labeled |
| $R_r \subseteq \mathcal{T}$ | Set of task indices that are at resolution $r$ |
| $G_{\text{task}}$ | Task dependency graph describing the correlation structure among tasks in a graph |
| $m$ | Number of sources |
| $\lambda_i$ | Output of $S_j$ for $X$, $\lambda_i \in \{-1, 1, 0\}$ |
| $\tau_j$ | Coverage set of $\lambda_j$ - the task indices $\tau_j \subseteq \mathcal{T}$ for which $S_j$ can label. For $S_j$ operating at resolution $r$, $\tau_j \subseteq R_r$ |
| $s_j$ | Size of the $j$th source coverage set, $s_j = |\tau_j|$ |
| $G_{\text{source}}$ | Source dependency graph that describe the correlation structure among source, particularly for correlations due to shared inputs |
| $G$ | Full dependency graph, $G = (V, E)$ obtained by combining $G_{\text{source}}$ and $G_{\text{task}}$. $V = \{\lambda_1, ..., \lambda_m\} \cup \{y_1, ..., y_{|\mathcal{T}|}\}$ |
| $\mu_j$ | Accuracy parameter for source $j$; $\mu_j(Z, W) = P(\lambda_j = Z \mid Y_{\tau_j} = W)$ |
| $\phi_{j,k}$ | Correlation parameter for sources $j, k$; $\phi_{j,k}(Z_1, Z_2, W) = P(\lambda_j = Z_1, \lambda_k = Z_2 \mid Y_\tau = W)$ |
| $\bar{P}_Y$ | Class prior for the $Y$ label vector |
| $a_j$ | Generalized agreement measure; $a_j(U, V) = \prod \lambda_j(U) \prod Y(V)$; Products are observable for common $V$ |
| $U, V$ | Subsets of the coverage set $\tau_j$ |
| $M$ | Matrix for first linear system, each row encodes pairs of agreements that factorize |
| $q$ | Observable vector with $\mathbb{E}[\lambda_j(U)\lambda_k(U')]$ terms |
| $\ell$ | Solution for products variable system system $M\ell = q$; Contains the terms $\log \mathbb{E}[a_k(U', V)]^2$ |
| $\rho_j$ | Product variable obtainable from generalized agreement; $\rho_j(U, V) = P(a_j(U, V) = 1) = \frac{1}{2} + \frac{1}{2}\mathbb{E}[a_j(U, V)]$ |
| $p_j$ | Joint distribution for source $j$ and $Y_{\tau_j}$; $p_j(Z, W) = P(\lambda_j = Z, Y_{\tau_j} = W)$ |
| $B_{2s_j}$ | Products-to-joints transformation matrix |
| $r$ | Vector containing the $\rho_j(U, V)$; is estimated after products variable system is solved |
| $e$ | Vector containing the $p_j(Z, W)$, solution to products-to-joints system $B_{2s_j}e = r$ |

Table 2: Glossary of variables and symbols used in this paper.

Next, we assume the result holds for $t = k$, and we show it is true for $t = k + 1$. That is, we have $B_t e_k = r_k$, and we'd like to show that $B_{t+1}e_{k+1} = r_{k+1}$. It follows from the definition of $B_{t+1}$ that it can be decomposed as

$$B_{t+1} = \begin{bmatrix} B_t & B_t \\ B_t & \bar{B}_t \end{bmatrix},$$

where the bar indicates flipped 1's and 0's. Furthermore, from our ordering, we have that $e_{k+1}$ can be written as $[e_k \cap (d_{t+1} = 1), e_k \cap (d_{t+1} = -1)]^T$, where we augment each probability term in $e_k$ with either $d_{t+1} = 1$ or $d_{t+1} = -1$. Similarly, we have that $r_{k+1} = [r_k; r_k \cup d_{t+1}]^T$. Then, what we want to show, $B_{t+1}e_{k+1} = r_{k+1}$, is equivalent to

$$\begin{bmatrix} B_t & B_t \\ B_t & \bar{B}_t \end{bmatrix} \begin{bmatrix} e_k \cap (d_{t+1} = 1) \\ e_k \cap (d_{t+1} = -1) \end{bmatrix} = \begin{bmatrix} r_k \\ r_k \cup d_{t+1} \end{bmatrix}.$$

The result follows almost immediately. For the block of $r_{k+1}$ on the top, we are summing the $B_t$ including both cases $d_{t+1} = 1$ and $d_{t+1} = -1$, which sums up to $r_k$ using the law of total probability and the inductive hypothesis. For the bottom block, we are summing over the probabilities of cases where $d_{t+1} = 1$ and the other terms in each $U$ multiply to 1, along with those with $d_{t+1} = -1$ and the

others multiplying to $-1$, which gives all the cases where the terms in $U\cup\{t+1\}$ multiply to 1, which is indeed the lower subvector on the right. Thus we are done. $\qquad\square$

Now we are ready for the proof of Theorem 1. Recall our assumptions: we see $n$ samples for each of the $m$ sources for sequences of length $T$, and we have the graph structure $G=(V,E)$. Our coverage sets are of length up to $T$. We have sufficiently many conditionally independent sources, and also that $\forall j, |\mathbb{E}[a_j(U,V)]| \ge b^*_{\min} > 0$. Finally, we assume that sign recovery is possible. One way to have this is to require that $\forall j, U, V, \sum_{\lambda_k \in S_j} \mathbb{E}[a_k(U,V)] > 0$ where $S_j$ is defined as in Algorithm 1.

With this, we prove a more general statement, without parameter reduction, and then we show how to obtain from it the parameter reduction case in Theorem 1.

**Theorem 2** *Under the assumptions previously described, let $\hat{\mu}$ and $\hat{\phi}$ be estimates of the true $\mu^*$ and $\phi^*$ produced with Algorithm 1. Then,*

$$\mathbb{E}[\|\hat{\mu}-\mu^*\|] \le 2^{2T}\sqrt{m}\,\frac{6}{p_{\min}b^*_{\min}}\|B_{2T}^{-1}\|\|M^\dagger\|\left(\sqrt{\frac{18\log(6\times 2^T)}{n}}+\frac{2\log(6\times 2^T)}{n}\right). \qquad (5)$$

*The expectation $\mathbb{E}\left[\left\|\hat{\phi}-\phi^*\right\|\right]$ satisfies the bound (4), replacing $\sqrt{m}$ with $m$, $B_{2T}$ with $B_{4T}$, and $2^T$ with $2^{2T}$.*

*Moreover, $\|M^\dagger\|=1$, and with parameter tying, (5) reduces to the expression in Theorem 1.*

**Proof:** There are two steps to the proof. First, we must show that the true parameters $\mu^*$ and $\phi^*$ are produced by Algorithm 1 when we have access to the true, population-level joint probabilities of the sources. Afterwards, we compute the noisy version due to sampling error.

**Population-Level Result** There are two necessary results: first, we need to show that the true parameters are solutions to the second system, and, secondly, that they are the unique solutions. We start with the first system. We work over each source $j$ and some fixed $U,V$ coverage sets for $\lambda_j$ and $Y$. From the algorithm, we have a set of sources $S_j \in V(G)$ with $c=|S_j\cup\{j\}|\ge 3$ so that $\lambda_j(U)$ and $\lambda_k(U_k)$ are independent conditioned on $\prod Y(V)$, and likewise for each pair of sources in $S_j$ over their corresponding $U$'s. For simplicity, we take $c=3$ exactly, but it is easy to solve larger systems, and the proof below does not depend on the value of $c$. We then say we have $S_j\cup\{j\}=\{j,k,f\}$, that is, our sources are $\lambda_j,\lambda_k,\lambda_f$.

We formulate the resulting matrix $M$. Recall that each row of $M$ corresponds to an equation

$$\log\mathbb{E}[a_j(U,V)]^2+\log\mathbb{E}[a_k(U',V)]^2=\log\mathbb{E}\left[\prod_{U,U'}\lambda_j(U)\lambda_k(U')\right]^2.$$

We have 3 such equations, for the pairs $(j,k),(j,f)$, and $(k,f)$. Then, our linear system is $M\ell=q$, given by

$$\begin{bmatrix}1 & 1 & 0\\1 & 0 & 1\\0 & 1 & 1\end{bmatrix}\begin{bmatrix}\log\mathbb{E}[a_j(U,V)]^2\\\log\mathbb{E}[a_k(U',V)]^2\\\log\mathbb{E}[a_f(U'',V)]^2\end{bmatrix}=\begin{bmatrix}\log\mathbb{E}\left[\prod_{U,U'}\lambda_j(U)\lambda_k(U')\right]^2\\\log\mathbb{E}\left[\prod_{U,U''}\lambda_j(U)\lambda_f(U'')\right]^2\\\log\mathbb{E}\left[\prod_{U',U''}\lambda_k(U')\lambda_f(U'')\right]^2\end{bmatrix}. \qquad (6)$$

The $3\times 3$ matrix above is full-rank, so that we can obtain the unique solution—the vector of $\mathbb{E}[a_j(U,V)]^2$ terms. Note that the above easily extends for more than 3 such equations. In the case of $c>3$ sources, the matrix has $\binom{c}{2}$ rows, each with exactly two 1's. The resulting matrix is also full-rank. To see this, we apply a result of A. M. Odlyzko [41] that states for 0/1, constant row-sum matrices (sum is 2 in our case), with $c>4$, $\binom{c-1}{2}+1$ distinct rows always guarantee that the matrix is full rank. For $c=4$, Odlyzko's result requires $2\binom{c-2}{(c-2)/2}+1=5$ distinct rows, and we have $\binom{4}{2}=6$, so this case works as well.

The only remaining step for the first system is to deal with identifiability. The above allows us to get the squares of the $\mathbb{E}[a_j(U,V)]$ terms. We thus need to recover their signs by using the sign recovery assumption. One obvious approach is to require that each of our sources has accuracy that satisfies

$\mathbb{E}[a_j(U,V)]>0$. However, much milder assumptions are possible: Note that once we know the sign of a single source accuracy, say $k$, we get all others, since for each source $f$, we have an equation for each pair $(k,f)$. So in fact, as mentioned in the assumptions, the much weaker requirement that $\sum_{\lambda_k \in S_j} \mathbb{E}[a_k(U,V)]>0$ is sufficient. There are other potential variants as well.

Now, by running the above procedure for all sources $j$ and all the $U,V$'s, we are ready to form the second system. We apply Lemma 1 with $t=2s_j$ and $D=(\lambda_j,Y_{\tau_j})$ obtaining that $B_{2s_j}e=r$.

To solve uniquely, we need to show that $B_{2s_j}$ is also full-rank. Note that the rank is not affected by adding a constant to each entry, unless it produces a 0 row, which it does not in this case, since the Hadamard matrix here was selected to have no all $-1$ rows. The Kronecker product of matrices multiplies the corresponding ranks. Since $\begin{bmatrix} 1 & 1 \\ 1 & -1 \end{bmatrix}$ is full-rank, $B_{2s_j}$ must be as well. Thus, there is a unique solution to our system, and it is indeed the desired $p_j(Z,W)$'s. Moreover, we can uniquely recover the $\mu$ parameters as well, as long as we know the distribution of $Y$. Finally, the same logic applies to the $\phi$ parameters, concluding the argument for the population-level result.

**Sampling-Level Result**    Now we apply a matrix concentration inequality to bound the sampling error. First, we require more notation.

For the first system, we'd like to estimate terms like $\log \mathbb{E}\left[\prod_{U,U'} \lambda_j(U)\lambda_k(U')\right]^2$. Again, say we are working with three sources $j,k,f$. Let us say they all work with the same coverage subset $U$ of maximal size $T$, which is an upper bound for our general case.

Then, for $j$ we create $o=2^T$ indicator variables, one for each configuration of $\lambda_j(U)$. Call these variables $c_{j,1},c_{j,2},...,c_{j,o}$, and likewise for $k$ and $l$. For example, if $T=2$, then $c_{j,1},...,c_{j,4}$ correspond to $\mathbb{1}\{\lambda_j=[-1,-1]\},\mathbb{1}\{\lambda_j=[-1,1]\}$, and so on.

We stack these vectors together to form the vector $c$ of length $3o$, and we estimate the matrix $O^*=\mathbb{E}\left[cc^T\right]$. We do this by estimating $c^1,c^2,...,c^n$ from our samples $\lambda^1,...,\lambda^n$, filling in the indicators accordingly. Then, we use the estimate $\hat{O}=\frac{1}{n}\sum_{i=1}^n c^i(c^i)^T$. Our first step is obtaining a bound on $\|\Delta_O\|=\|O^*-\hat{O}\|$.

We use the matrix Bernstein inequality following [53]. Let $\Delta_O = \hat{O} - O^* = \sum_{i=1}^n S_i$, where $S_i=\frac{1}{n}(c^i(c^i)^T-O^*)$. Then, using Theorem 1.6.2 in [53], we can write

$$\mathbb{E}[\|\Delta_O\|] \leq \sqrt{2v(\Delta_O)\log(6o)}+\frac{1}{3}L\log(6o). \tag{7}$$

Here, the dimensions of $\Delta_O$ are $3o \times 3o$, $v(\Delta_O)$ is the variance of $\Delta_O$, which is defined as $\|\mathbb{E}\left[\Delta_O\Delta_O^T\right]\|$, and, finally, $L$ is an upper bound on $\|\frac{1}{n}(c^i(c^i)^T-O^*)\|$. We can apply the result by taking the bound on $\|c^i\|^2$ to be $3o/n$ and a bound on $\|O^*\|$ to be $3o/n$ as well. We also need to bound the variance $v(\Delta_O)$; using the same ideas as in [53], we get $v(\Delta_O) \leq \frac{3o\|O\|}{n}$. Then, we have that

$$\mathbb{E}[\|\Delta_O\|] \leq \sqrt{\frac{18o^2\log(6o)}{n}}+\frac{2o\log(6o)}{n}. \tag{8}$$

This tells us how to bound the error between all the configurations that $\lambda_j$ and $\lambda_k$ can take on. We define $b^*$ as

$$b^* = \begin{bmatrix} \mathbb{E}\left[\prod_{U,U'} \lambda_j(U)\lambda_k(U')\right] \\ \mathbb{E}\left[\prod_{U,U'} \lambda_j(U)\lambda_f(U')\right] \\ \mathbb{E}\left[\prod_{U,U'} \lambda_k(U)\lambda_f(U')\right] \end{bmatrix}.$$

Note that the $U$ sets are the same for all the sources, but we're summing over all possible values for each pair.

We wish to bound $\|b^*-\hat{b}\|$, where $\hat{b}$ is the version of $b^*$ obtained with the use of the estimated $\hat{O}$. We do this for $j,k$ to write the following. By 'same sign' for $w,z$, we refer to them having the same parity

in the number of $+1$'s, so that the parity of the products of the terms agree.

$$|b^*_{j,k} - \hat{b}_{j,k}| = \left| \left( \sum_{w,z \text{ same sign}} P(\lambda_j = w, \lambda_k = z) - \sum_{w,z \text{ opp. sign}} P(\lambda_j = w, \lambda_k = z) \right) \right.$$
$$\left. - \left( \sum_{w,z \text{ same sign}} \hat{c}_{w,z} - \sum_{w,z \text{ opp. sign}} \hat{c}_{w,z} \right) \right|.$$

Here we broke up the product over the sum of terms that multiply to $1$ and those that multiply to $-1$. On the estimated side, we use the corresponding values of $\hat{c}$, which are our empirical estimates of the means of the $c$ indicators. Now, we upper bound by moving the sum out, to get

$$|b^*_{j,k} - \hat{b}_{j,k}| \le \sum_{w,z} |P(\lambda_j = w, \lambda_k = z) - \hat{O}_{wz}|.$$

Summing over all the sources, we get that

$$\|b^* - \hat{b}\|_1 \le \|O^* - O\|_1.$$

From this, we have that

$$\|b^* - \hat{b}\| \le \sqrt{3o}\|O^* - O\|. \tag{9}$$

Now we have control over the gap between $b$ and $b^*$. Recall that we form $\hat{q}$ from $\log(\hat{b}^2)$, then we solve the $3 \times 3$ system in (6). Let $\Delta_b = \hat{b} - b^*$. We have that, with the summation below running over the three pairs starting with $(j,k)$,

$$\|\hat{q} - q^*\|^2 = \sum_{(j,k)} \left( \log(\hat{b}^2_{j,k}) - \log((b^*)^2_{j,k}) \right)^2$$
$$= 4 \sum_{(j,k)} \left( \log(|\hat{b}_{j,k}|) - \log(|(b^*)_{j,k}|) \right)^2$$
$$= 4 \sum_{(j,k)} \left( \log(|b^*_{j,k} + (\Delta_b)_{j,k}|) - \log(|b^*_{j,k}|) \right)^2$$
$$= 4 \sum_{(j,k)} \left[ \log \left( 1 + \left| \frac{(\Delta_b)_{j,k}}{b^*_{j,k}} \right| \right) \right]^2$$
$$\le 4 \sum_{(j,k)} \left( \frac{|(\Delta_b)_{j,k}|}{|b^*_{j,k}|} \right)^2$$
$$\le \frac{4}{(b^*_{\min})^2} \sum_{(j,k)} (\Delta_b)^2_{j,k}.$$

Here we used the fact that $(\log(1+x))^2 \le x^2$. Next, we sum and take square roots and plug in our bound, (9)

$$\|\hat{q} - q^*\| \le \frac{2}{b^*_{\min}} \|\Delta_b\|$$
$$\le \frac{2\sqrt{3o}}{b^*_{\min}} \|O^* - O\|.$$

Next, we recall that $\hat{\rho} = \frac{1}{2} + \exp(\frac{\hat{\ell}}{2})$ and similarly for $\rho^*$. Here, the exponent is taken by entry. To obtain $\hat{\ell}$, we solve our system $M\hat{\ell} = \hat{q}$. Then, we have that

$$
\begin{aligned}
\|\hat{\rho} - \rho^*\| &= \left\| \exp\left(\frac{\hat{\ell}}{2}\right) - \exp\left(\frac{\ell^*}{2}\right) \right\| \\
&= \left\| \exp\left(\frac{\ell^*}{2}\right) \left( \exp\left(\frac{\hat{\ell} - \ell^*}{2}\right) - 1 \right) \right\| \\
&\leq \left\| \exp\left(\frac{\ell^*}{2}\right) \right\| \left\| \exp\left(\frac{\hat{\ell} - \ell^*}{2}\right) - 1 \right\| \\
&= \|\rho^*\| \left\| \exp\left(\frac{\hat{\ell} - \ell^*}{2}\right) - 1 \right\| \\
&\leq \sqrt{3} \left\| \exp\left(\frac{\hat{\ell} - \ell^*}{2}\right) - 1 \right\|.
\end{aligned}
$$

If $x$ is small, the we have that $\exp(x) \leq 2x + 1$. So, for large enough $n$, and thus the case of small $\hat{\ell} - \ell^*$,

$$
\left\| \exp\left(\frac{\hat{\ell} - \ell^*}{2}\right) - 1 \right\| \leq \|\hat{\ell} - \ell^*\|.
$$

Thus,

$$
\begin{aligned}
\|\hat{\rho} - \rho^*\| &\leq \sqrt{3}\|\hat{\ell} - \ell^*\| \\
&\leq \sqrt{3}\|M^\dagger\| \|\hat{q} - q^*\| \\
&\leq \frac{6\sqrt{o}}{b_{\min}^*} \|M^\dagger\| \|O^* - O\|.
\end{aligned}
$$

Since $\|x\|_\infty \leq \|x\|$, we also get that

$$
\|\hat{\rho} - \rho^*\|_\infty \leq \frac{6\sqrt{o}}{b_{\min}^*} \|M^\dagger\| \|O^* - O\|.
$$

This concludes our error analysis for the first system; we proceed to the second. Recall that we assemble the vector $r$ by stacking together $o = 2^T$ entries of various $\rho$'s. Thus,

$$
\begin{aligned}
\|r^* - \hat{r}\| &\leq \sqrt{o}\|\hat{\rho} - \rho^*\|_\infty \\
&\leq \frac{6o}{b_{\min}^*} \|M^\dagger\| \|O^* - O\|.
\end{aligned}
$$

Next, we deal with the second system: $B_{2T} e = r$ means $e = B_{2T}^{-1} r$, as $B_{2T}$ is full rank, so

$$
\begin{aligned}
\|\hat{e} - e^*\| &= \|B_{2T}^{-1}(\hat{r} - r^*)\| \\
&\leq \|B_{2T}^{-1}\| \|\hat{r} - r^*\| \\
&\leq \|B_{2T}^{-1}\| \frac{6o}{b_{\min}^*} \|M^\dagger\| \|O^* - O\|.
\end{aligned}
$$

All of this was for a fixed source, and we have $m$ such sources. Now, forming $\mu$ from the terms in $e$ only involves scaling by the probabilities of the $Y$'s; the smallest such term is $p_{\min}$. We have that

$$
\|\hat{\mu} - \mu^*\| \leq \sqrt{m} \|B_{2T}^{-1}\| \frac{6o}{p_{\min} b_{\min}^*} \|M^\dagger\| \|O^* - O\|.
$$

Taking expectations, we have that

$$\mathbb{E}[\|\hat{\mu} - \mu^*\|] \le \sqrt{m} \|B_{2T}^{-1}\| \frac{6o}{p_{\min} b_{\min}^*} \|M^\dagger\| \left( \sqrt{\frac{18o^2 \log(6o)}{n}} + \frac{2o \log(6o)}{n} \right).$$

Recall that $o = 2^T$, we get

$$\mathbb{E}[\|\hat{\mu} - \mu^*\|] \le 2^{2T} \sqrt{m} \frac{6}{p_{\min} b_{\min}^*} \|B_{2T}^{-1}\| \|M^\dagger\| \left( \sqrt{\frac{18 \log(6 \times 2^T)}{n}} + \frac{2 \log(6 \times 2^T)}{n} \right).$$

Now we have the general expression. In the case of parameter reduction, we take $T = 1$ in the powers, since all of our sources only use a single step, but now we use up to $mT$ of them, which replaces $m$. We then have,

$$\mathbb{E}[\|\hat{\mu} - \mu^*\|] \le \sqrt{mT} \frac{24}{p_{\min} b_{\min}^*} \|B_{2T}^{-1}\| \|M^\dagger\| \left( \sqrt{\frac{18 \log(12)}{n}} + \frac{2 \log(12)}{n} \right).$$

The expressions for $\phi$ follow similarly, but with pairs of edges, so that we replace $\sqrt{m}$ with $m$, and similarly in $T$. $\qquad\square$

## C.1 Identifiability

Next we discuss identifiability for our models. Our algorithm already implicitly guarantees identifiability under our assumptions, but we may be interested in which situations are sufficient, in the challenging multi-resolution setting with many latent labels, to guarantee model identifiability in general.

Our approach is to apply results from the seminal work [1], which provides identifiability results based on Kruskals' theorem on the uniqueness of 3-tensor decompositions. Applying this result in creative ways, [1] recovers results on identifiability for latent mixtures of product distributions, hidden Markov models (HMMs), and other latent variable models. The sense of identifiability here is *generic identifiability*, which corresponds to the information-geometric view. We consider the parameter space of our models as a variety, and demonstrate identifiability everywhere except potentially a measure-zero subvariety.

We break down our approach into two cases. The first case does not have any type of parameter reduction. The second case does. In both cases, we consider the parametrizations of both the true label and the sources. As usual, our sequence is of length $T$.

**General case**   First, we consider the case where the distribution of true labels does not factorize across time. We consider labels for all subsets of steps of lengths 1 ($Y_1$ resolution, or frames), length $T$ (the full-sequence level) and one additional resolution window level, such $g$, with $1 < g < T$. We assume that the labels are in $\{0, ..., r-1\}$, so that our label alphabet is

$$\mathcal{Y} = r^{T+1+(T-g+1)}.$$

Next, we consider $m$ sources that vote on (some part) of the time series sequence $X$. We view these as being independent conditioned on $Y$ (we can group non-independent sources together if necessary and simply count the remaining components).

Let us say that each supervision source is capable of producing one of $v$ possible votes on the time series. One concrete example is sources that can label each window of length up to $w$ (with each label having $r$ choices). Then the total number of possible votes is

$$v = r^T + r^{T-1} + ... + r^{T-w+1} = \frac{r^{T+1} - r^{T+1-w}}{r-1}.$$

Now we can apply Corollary 5 from [1], which states that identifiability is guaranteed if

$$m \ge 2 \lceil \log_v r^{2T+2-g} \rceil + 1.$$

For an example of how this works, consider $r = 2$, so that each subset gets a binary label, as throughout our paper. Then, $v = 2^{T+1} - 2^{T+1-w}$. If our window size is just 1, we get $v = 2^T$, and then, if, say $g = 2$, that $2 \lceil \log_v 2^{2T+2-g} \rceil + 1 = 2 \times 2 + 1 = 5$, so that we need 5 sources.

Figure 4: Simulation plots. (Left) Estimation error $||\hat{\mu} - \mu^*||^2$ decreases with increasing $N$ and improves with parameter tying (Ours). (Middle) Modeling sequential dependencies (Ours) leads to improved prediction performance over naive model. (Right) Our runtime is up to $90\times$ faster than Gibbs-sampling based approaches.

**Parameter Reduction** The previous approach is challenging practically, as we showed in our theorem in the previous section. The parameter space is very large—which makes parameter recovery challenging even if identifiability is ensured. In fact there is a tension between identifiability and recovery, since the first requires a large number of parameters, while the second is easier with fewer. Below we describe identifiability in the general case using based on parameter reduction. We exploit a type of reduction to the HMM model.

First, we use a Markovian model for $Y = (y_1,...,y_{2T+1})$. The $2T+1$ is simply for convenience here. We group together windows of length $u$. For convenience, say $u|(2T+1)$. Then, let each of the groups $(y_1,...,y_u),(y_{u+1},...,y_{2u}),...$ form a stationary Markov chain. The state space corresponds to the space of labels of subsets of the window, which has cardinality $r^{2^u}$—we allow labels for any particular subset of the window.

Next, we similarly set up a model for each of the sources. We set the number of outputs for each window of length $u$ to again be $v$. Again, if we have $m$ conditionally independent sources, our alphabet over the sources is $v^m$. Moreover, the Kruskal rank of the product distribution matrix of the sources has full Kruskal rank if each of the functions does. This puts us in a position to apply Theorem 6 from [1], for the HMM-style model we've just defined. We need, for identifiability, that the number of windows we label, which is $k = (2T+1)/u$, satisfies

$$\binom{k+v^m-1}{v^m-1} \geq r^{2^u}.$$

We can write this as a function of $T$:

$$\binom{(2T+1)/u+v^m-1}{v^m-1} \geq r^{2^u}.$$

Now we can express this in terms of particular variables; note that this is a tradeoff between:

- $u$, the complexity of the chain for $Y$,
- $T$, the parameter for the length of the sequence,
- $v$, the resolution of the supervision source votes, and
- $m$, the number of conditionally independent sources.

## C.2 Simulations

Simulations of effects of increasing $N$ and parameter tying on estimation error, effects of modeling sequential dependencies on prediction performance, and runtime are shown in Figure 4.

## C.3 Conditional Independencies

In Section 3.1, we discussed assumptions that are used in the parameter recovery algorithm. We described that we use a list of independencies among the source agreement measures (that is, the

$a_j(U,V)$ terms). These independencies can be derived from the source graphs, which are provided by the user. Below, we give additional details on this operation.

Note that there are multiple types of graphical models that provide us with such independencies. Our method is agnostic to this choice, as long as we can obtain the list of independencies. The illustrative example we consider is that of a binary Ising model where the sources have no singleton potentials. For simplicity of notation, we simply show a single label $Y = Y_1$ and the sources $\lambda_1,...,\lambda_m$ (that is, we take $\tau_1 = \{1,...,m\}$); normally, we would have a full model over all the $Y_i$'s and $\lambda_j$'s. We write the density as

$$f(Y,\lambda_1,\lambda_2,...,\lambda_m) = \frac{1}{Z}\exp\left(\theta_Y Y + \sum_{i=1}^{m}\theta_{Y,i}Y\lambda_i + \sum_{(i,j)\in E}\theta_{i,j}\lambda_i\lambda_j\right).$$

Here, $Z$ is the partition function, $E$ is the edge set among the sources in the graphical model, and the $\theta$'s are canonical parameters. The following argument provides the intuition for why the $a_j(U,V)$ are independent in this setting whenever their nodes are disconnected in the graph. We have that

$$f(Y,\lambda_1,\lambda_2,...,\lambda_m) = \frac{1}{Z}\exp\left(\theta_Y Y + \sum_{i=1}^{m}\theta_{Y,i}Y\lambda_i + \sum_{(i,j)\in E}\theta_{i,j}\lambda_i\lambda_j\right)$$
$$= \exp\left(\theta_Y Y + \sum_{i=1}^{m}\theta_{Y,i}Y\lambda_i + \sum_{(i,j)\in E}\theta_{i,j}(\lambda_i Y)(\lambda_j Y)\right)$$
$$= \exp\left(\theta_Y Y + \sum_{i=1}^{m}\theta_{Y,i}a_i(U,V) + \sum_{(i,j)\in E}\theta_{i,j}a_i(U,V)a_j(U,V)\right),$$

which indeed factorizes as long as there is no path in $G_{\text{source}}$ between $i$ and $j$ (other than the one through $Y$). Next, these terms are summed to produce a distribution over the $a_j$'s, and *symmetries* enable us to produce the desired independencies.

The above showed the simplified case where $U$ is a single source and $V$ consists of the $Y$ label only. This can be extended to the case where $|U| > 1$ or $|V| > 1$ (or both). There is a parity requirement for the symmetries to work. Specifically, at least one of the $a_i, a_j$ must involve an even number of terms, that is, $|U| + |V|$ is even.

## D    Extended Experimental Details

We describe additional details about the tasks described in Section 4, including details about supervision sources, the user-defined class prior, and the end model trained on labels generated by baselines and our method. Dataset statistics provided in Table 3.

### D.1    Dataset Details and Train/Dev/Test Splits

**Bicuspid Aortic Valive (BAV)**: We use the dataset from [13] and use the train/dev/test splits from that work.

**Interview Detection (Interview)**: We use the dataset from [15] and use the dev/test splits from that work. We additionally use an additional 57 hours of unlabelled data as the train split.

**Freezing Gait (Gait)**: Our dataset consists of sensor data sessions from different patients. We reserve a collection of sessions for dev and test, and split by patient to ensure that dev and test come from similar distributions.

**Movie Shot Detection (Shot)**: Our total dataset consists of 589 Hollywood movies (roughly 1200 hours). We treat windows of 16 consecutive frames as elements in our sequence notation and classify individual elements with the end model (so a sequence of length five is 80 consecutive frames). We have ground truth annotations for 45 minutes of data randomly distributed across 29 of the movies, which we split into the dev and test set by scene. We note that this gives an unfair advantage to traditional supervision, but `Dugong` outperforms nonetheless.

**ActivityNet Basketball Identification (Basketball)**: We take the subset of ActivityNet videos containing sports videos and aim to identify basketball videos. We sample one frame every two

seconds and classify individual frames as either coming from basketball videos or the other sports videos. We randomly select 5% of the videos as our dev set and 5% of the videos as our test set.

**Cyclist Detection in Self-Driving Car Dataset (Car)**: Our dataset consists of 50 minute of self-driving car dash cam footage, split into five videos [36]. The task is to identify whether individual frames, sampled at 10 FPS, contain cyclists. We select one video to split into dev and test, and reserve the rest of the videos as unlabelled training data. We split dev/test by taking a strided window of 5 frames with a stride of 10 frames (starting dev and test at frames 1 and 6 respectively) to ensure that dev and test come from the same distribution.

**Disorder Tagging in Electronic Health Record Text (EHR)**: This dataset consists of 299 patient notes sampled from MIMIC-III[26, 39], labeled for all mentions of disorders (e.g., aortic stenosis, pneumonia). The dataset is split into 133 development documents and 166 test documents, containing 10,940 and 16,641 sentences respectively. Training data consists of 10,000 sentences randomly sampled from 5,000 unlabeled MIMIC-III documents. Labels are generated per-word in IO (inside/outside) tag format.

## D.2    Task-Specific End Models

For BAV and Shot tasks, we use previously published end model architectures. For Gait and Open tasks, we rely on generic, off-the-shelf architectures commonly used for these modalities. We do not claim that these end models achieve the best possible performance for these tasks; our goal is to compare the relative improvements that our sequential weak supervision model provides compared to other baselines, which is orthogonal to achieving state-of-the-art performance for these specific tasks.

**Bicuspid Aortic Valve (BAV)**: We use the CNN-LSTM architecture described in [13] for use in classifying aortic valve malformations. This architecture includes a frame encoder for learning frame-level features and a sequence encoder for combining individual frames into a single feature vector. The frame encoder is a Dense Convolutional Network (DenseNet) [24] with 40 layers and a growth rate of 12, pretrained on 50,000 images from CIFAR-10 [31]. The sequence encoder is a bidirectional Long Short-term Memory (LSTM) [22] with soft attention [57]. All weights were fine-tuned during training. Models are trained using all MRI frames as input.

**Interview Detection (Interview)**: We use ResNet-50 pre-trained on ImageNet to classify individual frames of the video.

**Freezing Gait (Gait)**: We use a single layer bidirectional LSTM and hidden state dimension 300 as our end model that takes in a multivariate sensor stream as input. In order to provide longer sequential context, we pass in a windowed version of each candidate that includes past and future frames. Window size was tuned empirically, with [-3,+1] performing best overall. Since the sequence length of each frame slightly varies, we then pad these sequences (with 0's) and truncate any sequences over a pre-defined maximum sequence length. To provide more contextual signal, we also add multiplicative attention to pool over the hidden states in the LSTM.

**Movie Shot Detection (Shot)**: We use a C3D ConvNet with a ResNet-18 backbone pre-trained on the Kinetics dataset [19]. This is a common architecture for deep shot detection [20, 52]. We feed in 16 consecutive frames as input and classify whether or not there is a shot boundary in the 16 frames.

**ActivityNet Basketball Identification (Basketball)**: We use ResNet-18 pre-trained on ImageNet to classify individual frames of the video.

**Cyclist Detection in Self-Driving Car Dataset (Car)**: We only report label model results for this dataset, since we cannot release the proprietary end model used for this task.

**Disorder Tagging in Electronic Health Record Text (EHR)**: We only report label model results for this dataset.

## D.3    Supervision Sources

Supervision sources are expressed as Python functions with an average of 5 lines each. We list how many of the supervision sources operated on an element-level basis, a subsequence level basis (more than one frame), and a sequence level basis in Table 3. The supervision sources relied on the following information to assign noisy labels:

**Bicuspid Aortic Valve (BAV)**: First, the aortic valve in each frame was segmented using an intensity-based thresholding technique. The supervision sources relied on feature values derived from these segmented regions (i.e., area, perimeter, average intensity, eccentricity, and ratio of area and perimeter) to assign labels to each frame. Each supervision source assigned a label to the same frame that it used information from.

**Interview Detection (Interview)**: Two weak supervision sources use face identities on individual frames; they vote yes if Bernie Sanders or a host are detected in a particular frame, respectively. One labeling function checks whether the text "thank you" appears in the transcript within 30 seconds of a segment, and another checks whether there are the same number faces over the course of 30 seconds.

**Freezing Gait (Gait)**: The first supervision source employed uses stride time arrhythmicity [43, 44], which is calculated as average coefficient of variation for the past 3 stride times of the left and right leg. In addition to stride time arrhythmicity, other supervision sources we use involve the swing angular range of the shank, and the amplitude and variance in shank angular velocity. Out of the five total supervision sources used for this task, 3 of them operated on an element-level basis, and 2 of them operated on a sequence level (4 and 11 frames at a time).

**Movie Shot Detection (Shot)**: We compute frame-to-frame differences in HSV, RGB, and optical flow histograms. We detect frames that have large amounts of visual change from the frames immediately preceding them by detecting outliers in the frame-to-frame differences between histograms. These make up our three sequence-level weak supervision sources. We also introduce two weak supervision sources based on face detections. We run the MTCNN face detector [60] twice a second (once every twelve frames for a film shot at 24 FPS) and say that there is no shot change between detections if we find the same number of faces or if we find faces in the same location. On the other hand, if we find faces in different locations between detections, we say that there is a shot change. These make up our two subsequence-level weak supervision sources.

**ActivityNet Basketball Identification (Basketball)**: We use an off-the-shelf object detector [49] on one frame every two seconds to generate primitives. Our weak supervision sources operate on the objects detected in each frame; we detect whether a person and ball are detected in the frame, what the distance between the person and ball in the frame are, the color of the ball, and how much vertical distance the ball moves across a sequence.

**Cyclist Detection in Self-Driving Car Dataset (Car)**: We aim to classify whether frames contain cyclists in a representative sample of a self-driving car dataset. We detect whether frames have people or bicycles using an off-the-shelf object detector [37]. The object detect small bikes (i.e. when the bikes are far away), so we also write some heuristics for small person detections.

**Disorder Tagging in Electronic Health Record Text (EHR)**: Supervision sources are a collection of biomedical lexicons from the Unified Medical Language System (UMLS) [6] and a single stopword list. UMLS lexicons are broken down by semantic type (e.g., Disease or Syndrome, Finding) with each type mapped to a positive or negative label as per [14]. Positive and negative lexicons are merged by source vocabulary (e.g., SNOMEDCT_US) to generate 12 supervision sources.

| Dataset | End Model | T | $N_{train}$ | $N_{dev}$ | $N_{test}$ | m | $R_1$ | $R_2$ | $R_3$ |
|---|---|---|---|---|---|---|---|---|---|
| | | | Dataset Statistics | | | Supervision Statistics | | | |
| BAV | CNN-LSTM | 5 | 4329 | 10 6 | 94 | 5 | 5 | 0 | 0 |
| Interview | ResNet-50 | 5 | 6835 | 3026 | 3563 | 4 | 2 | 2 | 0 |
| Gait | LSTM | 3 | 1793 | 630 | 1014 | 5 | 3 | 0 | 2 |
| Shot | C3D ConvNet | 5 | 35,376 | 363 | 369 | 5 | 0 | 2 | 3 |
| Basketball | ResNet-18 | 5 | 3594 | 212 | 244 | 4 | 3 | 0 | 1 |
| Car | - | 5 | 4785 | 670 | 669 | 4 | 4 | 0 | 0 |
| EHR | - | 2 | 10,000 | 10,940 | 16,641 | 12 | 12 | 0 | 0 |

Table 3: We report the train/dev/test split in terms of the number of sequences in each set. The dev and test set have ground truth labels, which we assign labels to the train set using our method or one of the baseline methods. $m$ is the number of supervision sources, with $R_1$, $R_2$, and $R_3$ the number of supervision sources that label individual elements, subsequences, or the whole sequence, respectively.

### D.4 User-Defined Class Prior

We set the task-specific class prior for the tasks in the following manner. As discussed in Section 4, the user-defined prior outperformed the uniform and development set based for Open and Shot, but not for BAV and Gait.

**Bicuspid Aortic Valve (BAV)**: The labels were assigned on a sequence level for this task. The estimated incidence of BAV in the population is 1-2%, which was used to set the user-defined prior. However, the development and test sets have much higher prevalence rates (6-7%) as an artifact of their construction, thus an empirical prior derived directly from the development set performed best overall. Note the uniform class balance performs poorly due to incorrectly assuming that all label combinations within a sequence are equally likely, i.e., frames *within* a single sequence can alternate between BAV and normal.

**Interview Detection (Interview)**: The class priors were set starting with the class balance from the development set, and then slightly adjusting probabilities based on intuition of interview incidence.

**Freezing Gait (Gait)**: We design our own class prior by first starting with the class balance from the validation set, and then slightly adjusting probabilities based on intuition of freezing behavior. For example, we don't expect freezing and non-freezing behavior to alternate frequently in successive gait cycles so we assign very low probabilities to these events. Other sequences, such as consecutive freezing and consecutive non-freezing are likely more common, so we assign relatively high probabilities to these events.

**Movie Shot Detection (Shot)**: In this task, the labels are assigned to individual candidates in a five-candidate sequence. Each candidate is in turn a window of 16 consecutive frames. We set our prior based on the development set, but we manually reduce the likelihood of rare sequences to 0% (in particular, we set the likelihood of a sequence to 0% if we observe five or fewer instances in our development set).

**ActivityNet Basketball Identification (Basketball)**: In this task, labels are assigned per-frame for the end model, but our sequential modeling views sequences of five frames. We set our prior based on the development set, then slightly adjust the values based on intuition.

**Cyclist Detection in Self-Driving Car Dataset (Car)**: We set our prior based on the development set, but manually reduce the likelihood of rare sequences to 0 (in particular, we set the likelihood of a sequence to 0 if we observe five or fewer instances in our development set).

**Disorder Tagging in Electronic Health Record Text (EHR)**: We set our prior based on the development set.

### D.5 Detailed Results

We report detailed precision, recall, and F1 results for all datasets in Tables 4 and 5.

### D.6 Parameter Ablations

We examine how the following elements of our method improve empirical performance (Table 5):

*No parameter reduction (w.o Param Tie):* We force our model to learn a separate accuracy parameter per supervision source per resolution it labels and a separate correlation parameter per pairwise dependency. We show that this can hurt end model performance by 3.7 F1 points on average since there is not enough data to correctly estimate this many parameters.

*No sequential Dependencies (w.o Temp Deps):* We remove all sequential dependencies from our model, but still learn accuracy parameters for the supervision sources with parameter reduction. This hurts end model performance by 10.4 F1 points on average since removing the sequential dependencies among the supervision sources leads to "double counting" of the votes from sources that use similar information from the underlying data and overestimates the accuracies for these sources.

### D.7 Class Prior Ablations

We examine the effect of the user-defined prior for the distribution of labels in a sequence (Table 5):

*Uniform Probability (Uniform)*: All label configurations for a given sequence are equally likely.

*Prior based on Dev Set (Dev)*: Class priors are set empirically using the development set.

|  |  | Baselines | | | |
| --- | --- | --- | --- | --- | --- |
| Task | Metric | TS | MV | DP | Dugong |
| BAV | Precision | $26.1 \pm 3.8$ | $6.9 \pm 8.6$ | $70.0 \pm 19.8$ | $\mathbf{100.0 \pm 0.0}$ |
|  | Recall | $20.0 \pm 7.0$ | $5.7 \pm 7.0$ | $45.7 \pm 5.7$ | $\mathbf{37.1 \pm 7.0}$ |
|  | F1 | $22.1 \pm 5.1$ | $6.2 \pm 7.6$ | $53.2 \pm 4.4$ | $\mathbf{53.8 \pm 7.6}$ |
| Interview | Precision | $72.4 \pm 4.0$ | $48.7 \pm 5.7$ | $4.5 \pm 0.1$ | $\mathbf{89.6 \pm 4.2}$ |
|  | Recall | $89.5 \pm 3.0$ | $72.4 \pm 6.4$ | $\mathbf{99.1 \pm 0.0}$ | $94.6 \pm 0.8$ |
|  | F1 | $80.0 \pm 3.4$ | $58.0 \pm 5.3$ | $8.7 \pm 0.2$ | $\mathbf{92.0 \pm 2.2}$ |
| Gait | Precision | $65.2 \pm 13.7$ | $47.0 \pm 1.0$ | $50.3 \pm 1.6$ | $\mathbf{65.6 \pm 1.5}$ |
|  | Recall | $47.6 \pm 28.1$ | $\mathbf{89.8 \pm 3.0}$ | $84.1 \pm 2.2$ | $70.8 \pm 2.4$ |
|  | F1 | $47.5 \pm 14.9$ | $61.6 \pm 0.4$ | $62.9 \pm 0.6$ | $\mathbf{68.0 \pm 0.7}$ |
| Shot | Precision | $87.7 \pm 2.5$ | $79.7 \pm 2.1$ | $79.0 \pm 1.9$ | $\mathbf{87.8 \pm 2.9}$ |
|  | Recall | $79.3 \pm 1.3$ | $93.4 \pm 1.0$ | $\mathbf{94.3 \pm 1.2}$ | $87.6 \pm 3.4$ |
|  | F1 | $83.2 \pm 1.0$ | $86.0 \pm 0.9$ | $86.2 \pm 1.1$ | $\mathbf{87.7 \pm 1.0}$ |
| Basketball | Precision | $30.3 \pm 3.6$ | $10.0 \pm 6.9$ | $7.6 \pm 2.9$ | $\mathbf{33.0 \pm 4.0}$ |
|  | Recall | $24.1 \pm 0.4$ | $6.8 \pm 4.4$ | $8.0 \pm 3.7$ | $\mathbf{46.0 \pm 7.2}$ |
|  | F1 | $26.8 \pm 1.3$ | $8.1 \pm 5.4$ | $7.7 \pm 3.3$ | $\mathbf{38.2 \pm 4.1}$ |
| Car* | Precision | N/A | $57.7$ | $57.8$ | $\mathbf{95.3}$ |
|  | Recall | N/A | $81.3$ | $\mathbf{83.7}$ | $64.9$ |
|  | F1 | N/A | $67.5$ | $68.4$ | $\mathbf{77.3}$ |
| EHR* | Precision | N/A | $82.7$ | $79.1$ | $\mathbf{85.6}$ |
|  | Recall | N/A | $57.4$ | $61.2$ | $\mathbf{61.4}$ |
|  | F1 | N/A | $67.8$ | $69.0$ | $\mathbf{71.5}$ |

Table 4: Precision, recall, and F1 numbers for baselines. All reported values are means across five random weight initializations, $\pm$ standard deviation, except for the **Car** and **EHR** task, where we only report label model performance.

|  |  | Parameter Ablations | | Class Prior | | |
| --- | --- | --- | --- | --- | --- | --- |
| Task | Metric | w.o Param Tie | w.o Temp Deps | Uniform | Dev | User |
| BAV | Precision | $75.3 \pm 13.6$ | $42.5 \pm 17.0$ | $24.7 \pm 13.3$ | $\mathbf{100.0 \pm 0.0}$ | $99.8 \pm 0.5$ |
|  | Recall | $42.9 \pm 9.0$ | $\mathbf{45.7 \pm 5.7}$ | $17.1 \pm 5.7$ | $37.1 \pm 7.0$ | $34.3 \pm 14.6$ |
|  | F1 | $53.0 \pm 5.7$ | $41.9 \pm 6.7$ | $20.0 \pm 8.3$ | $\mathbf{53.8 \pm 7.6}$ | $48.1 \pm 14.5$ |
| Interview | Precision | $86.7 \pm 7.0$ | $80.8 \pm 5.3$ | $73.2 \pm 3.3$ | $\mathbf{89.6 \pm 4.2}$ | $88.4 \pm 3.4$ |
|  | Recall | $92.2 \pm 2.3$ | $88.4 \pm 7.3$ | $94.4 \pm 0.0$ | $\mathbf{94.6 \pm 0.8}$ | $91.4 \pm 6.8$ |
|  | F1 | $89.2 \pm 3.6$ | $84.2 \pm 3.8$ | $82.4 \pm 2.0$ | $\mathbf{92.0 \pm 2.2}$ | $89.8 \pm 4.9$ |
| Gait | Precision | $49.7 \pm 3.8$ | $66.5 \pm 2.1$ | $65.6 \pm 1.5$ | $57.9 \pm 2.9$ | $\mathbf{67.9 \pm 6.3}$ |
|  | Recall | $74.7 \pm 9.5$ | $64.7 \pm 5.7$ | $70.8 \pm 2.4$ | $\mathbf{80.5 \pm 3.7}$ | $66.3 \pm 7.5$ |
|  | F1 | $59.5 \pm 5.4$ | $65.3 \pm 2.2$ | $\mathbf{68.0 \pm 0.7}$ | $67.2 \pm 0.6$ | $66.3 \pm 1.1$ |
| Shot | Precision | $84.3 \pm 3.2$ | $82.3 \pm 1.8$ | $60.9 \pm 6.5$ | $70.8 \pm 2.3$ | $\mathbf{87.8 \pm 2.9}$ |
|  | Recall | $88.6 \pm 2.1$ | $91.4 \pm 1.1$ | $91.9 \pm 6.3$ | $\mathbf{95.0 \pm 1.1}$ | $87.6 \pm 3.4$ |
|  | F1 | $86.6 \pm 1.3$ | $86.6 \pm 0.5$ | $72.9 \pm 3.4$ | $81.1 \pm 1.2$ | $\mathbf{87.7 \pm 1.0}$ |
| Basketball | Precision | $24.2 \pm 3.4$ | $5.5 \pm 0.5$ | $10.0 \pm 0.0$ | $24.1 \pm 11.2$ | $\mathbf{33.0 \pm 4.0}$ |
|  | Recall | $\mathbf{51.6 \pm 9.0}$ | $45.1 \pm 4.1$ | $100.0 \pm 0.0$ | $33.5 \pm 22.8$ | $46.0 \pm 7.2$ |
|  | F1 | $32.9 \pm 4.9$ | $9.9 \pm 0.8$ | $18.3 \pm 0.0$ | $27.6 \pm 15.4$ | $\mathbf{38.2 \pm 4.1}$ |

Table 5: Precision, recall, and F1 numbers for ablations.

*User Defined Prior (User)*: The user defines a class distribution manually, and we provide task specific details in the Appendix. For Shot, the user-defined prior improves end model performance by 14.8 F1 points compared to uniform prior since shot transitions are rare events. Gait performs the best with a uniform prior, which is expected since there is no clear pattern in how freezing occurs while walking.