[Reviews · NeurIPS 2019]

Reviewer 1



The paper is sound and technically correct. There are contributions but I would not term them as significant. I'm not familiar with standard data sets in weak supervision for sequential data and thus I cannot assess the relevance and soundness of the computational study. The improvement is definitely significant, to the extent that it leans towards "too good to be true." I think algorithms based on graphical models (variational inference, Gibbs) should be added as benchmarks. The paper is very well written; easy to understand and without mistakes as far as I'm concerned. Based on the title I was expecting a stronger connection with classification. After all, the end goal is classification. Computational experiments are definitely about classification and there is the loss function bound stated in 'End Model Generalization.' However, this is a bound on loss and not generalization to unseen data. Besides, the bound is a direct consequence of Theorem. In summary, a true generalization bound would be much more significant.

Reviewer 2



The paper proposed one novel multi-resolution weak supervision problem for sequential data, which is quite interesting for both research and practical application. The problem formulation and key challenges are clearly demonstrated. For the proposed approach, the authors provide convincing explanations. Experimental resutls are also promising. For the weakness, see the following improvements suggestion.

Reviewer 3



Overall, the paper introduces an interesting algorithm with detailed theoretical and experimental analysis. My only comments are minor: - the introduction of the full set of untied parameters in 2.2 then tying them in 2.3 seems cumbersome and a bit confusing on the first read - could the tied parameters be introduced directly instead? - line 147 references section 3.1.4 but there is no such section in the paper Update: The paper discusses the advantages of the proposed algorithm over Gibbs sampling approaches in terms of speed and convergence, and compares against a Gibbs baseline without sequential correlations, but it seems it should also be possible to create a Gibbs-based model that does consider sequential correlations. It would be interesting to see a comparison against this stronger baseline in terms of empirical task performance.

[Author Response · NeurIPS 2019]

We appreciate all of the reviewers' valuable comments and feedback, which have helped improve our paper. We will shortly release a PyTorch implementation with tutorials and detailed commands for easy reproducibility. We respond to each of the reviewers' comments below.

**Reviewer 1.** We begin by addressing R1's concerns about the impact and importance of our work, including whether we are the first to study weak supervision for sequential data. First, we agree with the reviewer that the related work section is short, and will edit the draft to add references related to existing weak supervision and sequential modeling works. None of the existing methods in weak supervision handle *multi-resolution* sources over *sequential* data.

We find that multi-resolution labels for large-scale data applications like video are critical to performance gains. This key notion motivates our work. We highlight this idea with several real-world tasks from the draft:

(1) We collaborated with a *large self-driving car company* using our method to label their video data to train models for autonomous driving. The **Car** task (line 268-269, Figure 3, Table 4 in Appendix) is a sampled version of this task.

(2) We *improved over results recently published in Nature Communications* related to labeling MRI video data at the population scale (4000 patients, Table 3). We report these results as the **BAV** task (line 253-355, Table 1), which improves over published state-of-the-art results by 5.7 F1 points.

(3) We demonstrated how our method can be used for textual BIO (beginning-inside-outside) tagging via the **EHR** task (lines 689-694, Table 3 in Appendix). This task operates over real-world patient data, and our method is able to *model a collection of lexicons from a public database* as multi-resolution supervision sources (lines 765-769 in Appendix).

Since submission, we are also working with a *large video-sharing website* to label content across millions of videos.

Third, R1 asks about the generalization bounds and the broader impact of our theoretical results. We indeed prove a generalization bound for classification problems in our work; specifically, we bound the expected loss of our model on a random unseen data point, as is standard in generalization theory. Note that the loss function in our bound is flexible, and could correspond to several losses. The result is a first step towards future results that more finely characterize generalization with no ground truth labels. The result and the theorem it is based on has broader implications beyond just our algorithm; for example, the argument of line 222-223 shows that parameter sharing is critical to efficiently modeling temporal data (i.e., without sharing, the number of samples scales exponentially in the sequence size). We will further clarify these notions in the updated draft.

Runtime vs. N,m

Finally, R1 asks about baseline benchmarks. We compare to Gibbs sampling in Section 4, Table 1 when we improve over data programming (DP) [45], which is a Gibbs-sampling based weak supervision method. We also compare to Gibbs in terms of accuracy (Appendix, Figure 4, middle) and runtime (right). We find a $90\times$ improvement in runtime — a significant speedup. Variational methods did not scale up sufficiently to handle our setting (large amounts of data and no labels). To better understand the gains attained by our method relative to baselines, we have included a qualitative analysis in Figure 3 of our submission. For reproducibility, we included our code with a README in the supplementary materials.

**Reviewers 2 and 5.** We agree with and appreciate R2 and R5's suggestions about improving the writing and presentation style. Since submission, we have improved and revised our presentation by simplifying notation, adding intuition, and providing more examples. R5 comments on the presentation of parameter tying. We have edited the draft accordingly to present the concept more directly, leaving the most general case to the appendix. We will also correct the reference to the non-existent section in the revised draft.

R2 asks whether our method is applicable in cases where a single source assigns labels at multiple resolutions. Our method can handle this situation. We explain using a concrete example. Say we have a weak supervision source that assigns labels at the frame-level ($F$) and at the scene-level ($S$). Then, the output vector for the source would be

$$Y = [\text{F}^1_{label}, \text{F}^2_{label}, \text{F}^3_{label}, \text{S}_{label}],$$

where $\text{F}^i_{label}$ refers to the label for the $i$-th frame in the scene and $\text{S}_{label}$ refers to the label at the scene-level.

For a datapoint where the source only assigns frame-level labels (say all positive), the output vector would look like $Y = [1, 1, 1, 0]$, where 0 is an abstain. In case it applies a label to $F^1$ and the overall scene, it would look like $Y = [1, 0, 0, 1]$. The generative model would take into account the hierarchical nature of the labels, since the structure is encoded in the model (Figure 1 in the draft). The output vector of the generative model would have the same format with a probabilistic label per entry (for frames $F^1$, $F^2$, $F^3$, and scene $S$ in this case). The user can then choose the resolution at which she requires training labels. We will edit our draft to include clarifications about and add examples for how a single source can label at multiple resolutions.

[Meta-Review · NeurIPS 2019]

The authors present an algorithm for leveraging noisy "weak" labels at multiple resolutions (e.g., frames/scenes/full-video) taking into account the sequential nature of labels as well as the correlation between labels in order to produce a combined label that is then used to train a downstream model. The authors provide theoretical guarantees that bound the convergence rate of the combining model parameters under reasonable conditions. This convergence rate can then be used in a standard way to guarantee the generalization of the downstream algorithm. Finally, the authors also present extensive experiments on real-world datasets where multi-resolution labeling could reasonably be used and show superior performance over three baseline methods, including Data Programming which the authors claim is state-of-the-art. I along with the reviewers agree that the design and analysis of the algorithm provides a significant contribution and that the empirical results demonstrate effectiveness. I recommend for acceptance.